

# How to simulate radiative inputs in complex topographic areas, an analysis on 115 Swiss Alps weather stations

Philippe Riboust[1,2], Nicolas Le Moine[1], Guillaume Thirel[2], and Pierre Ribstein[1]

[1]Sorbonne Universites, UPMC Univ. Paris 06, CNRS, EPHE, UMR 7619 Metis, 4 place Jussieu, 75005 PARIS, FRANCE
[2]Hydrosystems and Bioprocesses Research Unit (HBAN), Irstea, 1, rue Pierre-Gilles de Gennes, CS 10030, 92761 Antony Cedex, France

*Correspondence to:* Philippe Riboust (philippe.riboust@upmc.fr)

**Abstract.** In hydrology, solving the energy balance could be needed to estimate evapotranspiration or snowmelt. Shortwave (SW) and longwave (LW) incoming radiation fluxes are often derived from climate models or reanalyses. The resolution of these models is often too scarce to reproduce the variability of the energy fluxes in complex topography situations. For hydrological purposes, these radiations can also be simulated using empirical formulations with only maximum and minimum daily air temperature data as inputs. Many different formulations have been developed, mainly for specific areas. In this paper, we modify the existing Bristow atmospheric transmissivity equation so that it can be adapted to different areas and different elevation ranges. A calibration method for the transmissivity equation, coupled to the Sicart emissivity parameterization, has been determined. The new parameterization was developed and validated on a total of 115 meteorological stations from the MeteoSwiss network over the Swiss Alps. The results showed an increase in performances for simulating shortwave radiations in high-elevation areas, without decreasing the performance for lower elevations. However, the performance increase on transmissivity did not improve the longwave simulations as had been expected. The parameterization was also validated on reference surface temperature by solving a simplified energy balance. This makes it possible to evaluate the performance of the two parameterizations combined, considering a typical retro-action of the snow surface. This provided good results with a slight increase in performance compared to the original formulation. It also showed that the inaccuracy of SW radiations has a greater impact on the performance of the reference surface temperature than LW radiation errors. The drawback of this new formulation is that it converges to unexpected parameter values at the calibration process, which requires setting some parameters before calibration.

## 1  Introduction

Many fluxes need to be measured to reconstruct the whole energy balance, but unfortunately there are very few places in the world where all the necessary fluxes are measured simultaneously. Raleigh et al. (2015) observed that 99% of US weather stations do not measure all the data needed to force physically-based hydrological models, the least measured being turbulent fluxes and longwave radiation data. In addition, few meteorological stations are located in mountainous areas, especially at high elevations (Dettinger, 2014; Le Moine et al., 2015; Valéry et al., 2010). As physically-based hydrological models often



use energy flux to drive evapotranspiration and snowmelt processes, this lack of meteorological stations induces uncertainties in hydrological models simulations.

To overcome data scarcity, energy balance models often use forcings coming from atmospheric models and reanalyses (Shrestha et al., 2006; Vionnet et al., 2012). Often reanalysis outputs are too scarce for mountain areas and they need to be

downscaled in order to take into consideration the complexity of the topography, which can lead to a number of uncertainties. Another way to compute the energy balance is to empirically model every energy balance component, as done by Archibald and Walter (2014) for evapotranspiration modeling and by Walter et al. (2005) for snow modeling. Using parameterizations to determine missing incoming shortwave (SW) and longwave (LW) radiation data has the advantage of requiring only very few data: only daily temperature range data, which are widely available, are required.

To calculate SW and LW radiations using only widely measured data (ie., temperature and/or precipitation), it is necessary to parameterize atmospheric transmissivity ($\tau$), which indicates how the atmosphere absorbs or reflects the incoming SW radiations, and atmospheric emissivity ($\epsilon$), which indicates the ability of the atmosphere to emit LW radiation (compared to the black body model). Atmospheric transmissivity ($\tau$) depends on the air mass crossed by the sun's rays, the cloud cover and the concentration of water vapor and aerosol particles (Shook and Pomeroy, 2011). Atmospheric emissivity varies with air

composition, especially with the water vapor content and cloud cover (Conway et al., 2014).

Besharat et al. (2013) reviewed 78 different transmissivity models, which can be classified into three main types: diurnal temperature range models, sunshine duration models, and cloud cover models, the second giving the best results but also using the least available type of data. The best known models are those designed by Hargreaves and Samani (1982), Bristow and Campbell (1984) and Thornton and Running (1999), which are diurnal temperature range models.

Many different emissivity parameterizations have also been created and tested in the literature (Ebrahimi and Marshall, 2015; Sicart et al., 2006, 2010). Juszak and Pellicciotti (2013) tested 13 clear-sky emissivity parameterizations in Switzerland and seven different cloud cover corrections. They found that spatial and seasonal calibrations are necessary for modeling LW radiations. Gubler et al. (2012) also found that parameter calibration is necessary for simulating SW and LW radiations at multiple sites in Switzerland and that the parameters depend strongly on the elevation of the station considered. These

studies show that the parameter values of the transmissivity and emissivity formulations are dependent on time and location variables, thus making it hard to propose universal parameter values. This is also reinforced by the fact that a high number of parameterizations have been developed for specific areas and study sites.

The snowpack has a direct response to these radiative forcings as some of the incoming SW radiations are reflected and LW radiations are emitted by the snowpack. Therefore, it is difficult to estimate the uncertainties of SW and LW simulation

on the whole system without modeling the other components of the energy balance. To overcome this problem, multiple studies validated the simulated SW and LW radiations on mountainous areas using an energy balance snow model (Juszak and Pellicciotti, 2013; Mölg et al., 2009; Pellicciotti et al., 2011). Lapo et al. (2015) used the UEB snow model to quantify the impact of inaccuracies in incoming radiation simulations and showed that bias errors on longwave radiations have a greater impact on the snow water equivalent (SWE) simulated than the standard deviation of random noise added to the signal. Thus,





increasing the performance of SW and LW radiations does not guarantee a decrease of the overall inaccuracies transmitted to the energy balance models.

The objective of the present study is to determine a better way to estimate incoming SW and LW radiations in mountainous areas using only limited data inputs. This is done by creating a generic parameterization of SW radiations, namely using daily temperature range data only, with adaptive parameters that must be calibrated at the mountain range spatial scale. To overcome the errors in the SW and LW simulations due to the complex topography, the dependency of existing parameterizations with the elevation and an adequate calibration strategy for the radiation model will be investigated and assessed. Finally, the results will be discussed and the simulated SW and LW radiations will be validated on a reference snow surface temperature. The simple method that will be introduced in this article will give an overview of the errors that should be expected using the simulated SW and LW radiations to drive an energy balance snow model.

## 2  Data sets

This study was conducted on 115 automatic weather stations located in Switzerland, which are managed by MeteoSwiss (https://gate.meteoswiss.ch/idaweb/). These stations record at least air temperature, precipitation and vapor pressure at a daily time step. SW incident radiations are measured at almost all weather stations (108), while only some of them also include LW radiation measurements (a total of 33 stations). Only one station from this data set measures LW radiations without measuring SW radiations. The 115 stations are well distributed across the country (Fig. 1), with $60\%$ of the stations located at less than 1000 m elevation, $25\%$ of stations located between 1000 and $2000\,m$ elevation and $15\%$ above $2000\,m$ elevation. All stations have at least 1 year of measured data. If available, data were collected from 2009 to 2015.

In addition, a 100 $m$-resolution digital elevation model (DEM) of Switzerland was used in this paper. A comparison between observed and simulated SW and LW radiations will be made for three specific stations: the Koppigen (KOP, $484\,m$), Samedan (SAM, $1708\,m$) and Jungfraujoch (JUN, $3580\,m$). Their location is indicated in Figure 1. This set of stations is considered as a representative sample of the differences in energy balance components between different elevations of the Swiss Alps.

[FIGURE 1]

## 3  Parameterization development

### 3.1  Parameterization of shortwave radiations

The incoming azimutal flux from the Sun at the top of the atmosphere is at a constant $1360\,W.m^{-2}$. However, a unit surface at the Earth's surface only receives part of this energy since it depends on the angle of incidence of the solar rays and on how much the atmosphere reflects and absorbs the incoming solar energy. The equation of the incoming global solar radiation $R_g$, which takes into account direct and diffuse SW radiations, can be written as:

$$R_g = \tau\, R_{pot} \tag{1}$$





with $R_{pot}$ $(W.m^{-2})$ the theoretical potential SW radiation with no attenuation of the solar radiation by the atmosphere and $\tau$ represents the atmospheric transmissivity. $R_{pot}$ can be simply computed knowing the Julian day, the latitude, the slope inclination, and the orientation of the surface considered (Sproul, 2007). $R_{pot}$ can be computed on an entire DEM using a radiation model such as the one proposed by Suri and Hofierka (2004) in GRASS GIS. This method computes the potential daily radiations taking the surrounding topographic shadowing into account. This is advantageous mainly at mountainous sites where the topographic shadowing can play an important role in simulating snow and glacier melt (Gao et al., 2017).

One of the best known parameterizations of atmospheric transmissivity using the daily temperature range comes from Bristow and Campbell (1984) and is written as:

$$\tau = A(1 - exp\left(-B\Delta T^C\right)) \tag{2}$$

$$B = 0.036\, exp(-0.154\, \overline{\Delta T_m}) \tag{3}$$

where A represents the maximum transmissivity, which was fixed by Bristow at a value of 0.70, C is a constant equal to 2.4 (A, B and C without units) and $\overline{\Delta T_m}$ is the monthly mean daily temperature range (in $K$). We rewrote the Bristow formulation (Eq. 2 and Eq. 3), with more explicit parameter names and with the introduction of time and space dependency variables:

$$\tau(t, M) = \tau_{max}(M)\left[1 - exp\left(-\frac{\Delta T(t, M)}{\Delta T_{ref}(M)}\right)\right] \tag{4}$$

The formulation of transmissivity is not only dependent on time ($t$), but also on the location of the area studied ($M$). The position $M$ is defined by $x$ and $y$ coordinates and its elevation $z$ ($M(x, y, z)$). The power function (parameter $C$ in Eq. 2) was disregarded in order to decrease the number of parameters, given that simplifying it shows little difference in transmissivity simulations (Pellicciotti et al., 2011). The objective of the following sections is to set a parameterization for the maximum transmissivity parameter $\tau_{max}(M)$ ($A$ in Eq. 2) and a parameterization for the reference parameter for the daily temperature range $\Delta T_{ref}(M)$ ($1/B$ in Eq. 2) to regionalize the Bristow formulation across different elevation ranges.

### 3.1.1 Parameterization of the maximum transmissivity

The maximum transmissivity that can be reached at a given location, ie., when the sky is completely clear, depends on the air mass and on the presence of aerosol particles. This air mass can be characterized by the atmospheric pressure and is clearly related to the elevation of the study area. Knowing this, the dependency of the $\tau_{max}$ parameter on the space location was simplified taking into account only the elevation component ($z$). To take this into account, we introduced the following formulation:

$$\tau_{max}(z) = 1 - [1 - \tau_{max}(z = 0)]\, exp\left(-\frac{z}{z_{ref}}\right) \tag{5}$$





Here, the maximum transmissivity achievable at an elevation z depends on two parameters: the maximum transmissivity at an elevation of 0 meters above sea level (m) $\tau_{max}(z=0)$ and an elevation parameter $z_{ref}$ $(m)$ determining how fast the value of $\tau_{max}$ evolves with an increase of elevation. This formulation has the advantage of being naturally bounded between the $\tau_{max}(z=0)$ value and 1. The formulation was fitted onto the observed $\tau_{max}$ values (Fig. 2). To avoid introducing errors using

5 the maximum transmissivity on measured SW radiation, the observed maximum transmissivity was taken for each station as equal to the 95th percentile of the observed transmissivity distribution.

[FIGURE 2]

The results show a good agreement of the simulated maximum transmissivity with the elevation compared to the observations ($R^2 = 0.71$). The fitting of this parameterization gave $\tau_{max}(z=0) = 0.75$ and $z_{ref} = 2000\ m$.

10 **3.1.2 Parameterization of the daily temperature range reference**

As for the maximum transmissivity parameterization, we wish to parameterize the $\Delta T_{ref}(M)$ descriptor in order to be dependent on the location of the study area. Analyzing the mean annual daily temperature range (Fig. 3) shows that the annual temperature range regime is variable across the different meteorological stations.

[FIGURE 3]

15 This shows that the parameterization of transmissivity with the daily temperature range has to take into account that the $\Delta T_{ref}$ parameter should be variable depending on the location. This is taken into account in the Bristow formulation by the $B$ parameter, which varies depending on the mean monthly daily temperature range observations ($\overline{\Delta T_m}$). We chose a first formulation (Eq. 6) to simplify the $B$ parameter from Bristow:

$$\Delta T_{ref}(M) = k\overline{\Delta T_a}(M) \tag{6}$$

20 Here, the $\Delta T_{ref}$ parameter depends only on a multiplicative coefficient $k$ and on the mean annual daily temperature range $\overline{\Delta T_a}(M)$ at the desired station. A second method was developed to parameterize $\Delta T_{ref}$ using only topographic data, because it seems that the mean annual $\overline{\Delta T_a}$ is dependent on topography. Looking at Figure 3, the relation between the mean annual daily temperature range is not only related to the absolute elevation of the meteorological station because a linear correlation is difficult to see. This difference could be related to the morphology of the surrounding topography of the area studied (ie., in

25 a valley, at a peak, at a flat area). In valleys, the presence of anabatic and katabatic winds tends to increase the average daily temperature range, when in contrast, mountain peaks tend to have a low daily temperature range, since they are more impacted by synoptic meteorology than by the local radiative forcing (Carrega, 1995).

This topographic morphology was characterized using a buffer around the desired station and studying the neighboring pixel elevations around the station considered. We named $\delta$ the differences in elevation (Eq. 7).

30 $$\delta = z - z_s \tag{7}$$





Examining the distribution of the $\delta$ values for three stations with a 2000 $m$ buffer gives the following Figure 4.

[FIGURE 4]

Different descriptors were tested by fitting the parameterization to the mean annual temperature range observed. We found that the absolute elevation of the area considered $z$ $(m)$ and the average elevation of the distribution noted $\overline{\delta}$ $(m)$ are the best descriptors. The average value of the differences in elevation $\delta$ gives the following information about the location of the meteorological station:

– If the average is negative (positive), this means that the station is at a higher (lower) elevation compared to the average of the distribution.

– If the absolute value of the average is high, this means that the difference in elevation between the station and its surroundings is high. A highly negative average value means that the area considered is at a peak and a highly positive value means that the area considered is in a valley.

– If the absolute value of the average is low, this means that either the difference in elevation between the station and its surroundings is small or that the station elevation is the same than the average elevation as the distribution. It means that the station is in a flat area or that the station is in the middle of a slope.

Using $z$ and $\overline{\delta}$ to parameterize the $\Delta T_{ref}$ parameter gave the following equation:

$$\Delta T_{ref}(M) = \Delta T_{param} exp\left(\frac{\overline{\delta}}{\delta_{ref}} - \frac{z}{z_{ref}}\right) \tag{8}$$

where

$$\delta_{ref} = R\,m_{ref} \tag{9}$$

This parameterization depends on three parameters. The first one is a temperature parameter $\Delta T_{param}$ $(K)$, which can be equated with the average daily temperature range for a station in a flat area at sea level ($\Delta T_{param} = \Delta T_{ref}(z = 0; \overline{\delta} = 0)$). The $z_{ref}$ $(m)$ parameter is similar to the parameter presented in the previous section (Eq. 5). Given, that the range of $\overline{\delta}$ values greatly depends on the radius of the buffer $R$ that is taken to calculate the distribution, it was decided to define the $\delta_{ref}$ parameter as being equal to the radius of the buffer $R$ $(m)$ times a slope parameter $m_{ref}$ $(-)$ (Eq. 9). The $m_{ref}$ value stays relatively stable for different $R$ values compared to $\delta_{ref}$, making it easier to calibrate. To study the effect of the $\overline{\delta}$ variable, a simplified formula using only $z$ (Eq. 10) and a second using only $\overline{\delta}$ (Eq. 11) were also tested:

$$\Delta T_{ref}(z) = \Delta T_{param} exp\left(-\frac{z}{z_{ref}}\right) \tag{10}$$

$$\Delta T_{ref}(M) = \Delta T_{param} exp\left(\frac{\overline{\delta}}{\delta_{ref}}\right) \tag{11}$$





The summary of the different $\Delta T_{ref}$ parameterizations used in this study (Eqs. 6, 8, 10 and 11) is available in Table 1.

[TABLE 1]

The influence of the radius $R$ on the parameterizations of $\Delta T_{ref}$ was studied by using the following equation:

$$\Delta T_{ref}(M) = \overline{\Delta T_a}(M); \; if \; \Delta T_{param} = \overline{\Delta T_a}(z=0, \overline{\delta}=0) \tag{12}$$

This makes it possible to find values for $m_{ref}$ and $z_{ref}$ parameters using the mean annual temperature range values observed for each station. The results in Figure 5a show the RMSE values associated on the three equations for different buffer $R$ values.

[FIGURE 5]

Figure 5a shows that using a value of 2000 m for $R$ seems to be the best compromise for minimizing the model error with a limited increase in calculation. Also, the best RMSE performance (1.1 K) is found using the $\Delta T_{ref}(t, M)$ parameterization,

the $\Delta T_{ref}(t, z)$ having a RMSE value of 1.2 K and the $\Delta T_{ref}(t, \delta)$ having a RMSE value of 1.5 K for a buffer of 2000 m. As the Eq. 8 gives better performance than the other two formulations, this parameterization was chosen for the computing the results below.

Figure 5b shows an example of how the $\overline{\delta}$ variable can be used to identify peak, valley or flat regions. A weather station was considered at a peak when the $\Delta T_{ref}$ associated was below 90% of the $\Delta T_{param}$ value. On the other hand, stations were

considered to be located in valleys when the ratio between $\Delta T_{ref}$ and $\Delta T_{param}$ was above 110%. Overall, it seems that using an adequate parameter $\delta_{ref}$ distributes the stations advantageously. Some errors can be seen, mainly for stations located in valleys, but the partitioning is adequate on the whole. The results of the $\Delta T_{ref}$ parameterization are available in Figures 5c and 5d. They show a fairly good simulation of the mean annual daily temperature range for almost all the stations using Eq. 8. Nevertheless, stations with very high observed daily temperature range are clearly underestimated from 1 to 2 K, resulting in

an $R^2$ value of 0.62. The regression gave 0.544 and 8180 $m$, respectively, for the $m_{ref}$ and $z_{ref}$ parameters of Eq. 8 with an $R$ value of 2000 $m$.

The summary of the transmissivity models used in this study is available in Table 2:

[TABLE 2]

### 3.2    Parameterization of longwave radiations

The other part of the incoming radiation budget is the LW radiation emitted by the atmosphere. It can be calculated using the Stefan-Boltzmann law, which is based on the black body theory:

$$R_{lw\downarrow} = \sigma \, \epsilon \, T^4 \tag{13}$$

The amount of radiation ($R_{lw\downarrow}$) emitted by an object only depends on its temperature ($T$ in K), its emissivity property ($\epsilon, -$) and the Stephan-Boltzmann constant ($\sigma$), which is equal to $5.67.10^{-8} \; W.m^{-2}.K^{-4}$. Atmospheric longwave radiation depends

mostly on the lower layers of the atmosphere the air temperature can be approximated by measured near surface air temperature (Ohmura, 2001).





The atmospheric emissivity was parameterized by Brutsaert (1975) for clear sky (without any cloud cover) conditions using the following equation:

$$\epsilon_{clear}(t,z) = \epsilon_1 \left( \frac{e(t,z)}{T(t,z)} \right)^{1/\epsilon_2} \tag{14}$$

where $T$ defines the atmospheric temperature ($K$) and $e$ represents the vapor pressure ($hPa$). $\epsilon_1$ and $\epsilon_2$ are two parameters that were fixed by Brutsaert to the values of $1.24$ $(-)$ and $7$ $(-)$, respectively. This equation is commonly known to give good results (Herrero and Polo, 2012; Juszak and Pellicciotti, 2013), but it cannot be used if the sky is cloudy. Based on this clear-sky emissivity model, many authors defined their own parameterization in order to take cloud cover into account. In 1982, Brutsaert improved his clear-sky formulation for an all-sky parameterization using cloud cover data as a variable (Brutsaert, 1982). Sicart et al. (2006, 2010) created two different formulations for calculating emissivity in mountainous regions, one for the Alps using relative humidity ($RH$) and transmissivity as input variables (Sicart et al., 2006) and another more specific to tropical regions, using transmissivity only (Sicart et al., 2010). These formulations are still based on the Brutsaert clear-sky model but use different approaches to take cloud cover into account. The Sicart (2006) formula was selected for this study:

$$\epsilon(t,z) = \epsilon_{clear}(t,z) \left( 1 + \frac{RH(t,z)}{RH_{ref}} - \frac{\tau(t,z)}{\tau_{ref}} \right) \tag{15}$$

This parameterization of the cloudiness effects adds two parameters the reference relative humidity $RH_{ref}$ $(-)$ and the reference transmissivity $\tau_{ref}$ $(-)$. This formulation depends on the previously parameterized transmissivity, and it was therefore assumed that an improved transmissivity parameterization should also improve the simulation of emissivity at higher elevations. Atmospheric humidity is a variable needed to compute atmospheric emissivity. Since the objective of this paper is to simulate both SW and LW radiations only with air temperature, a parameterization of humidity has to be used.

### 3.3 Humidity parameterization

The vapor pressure and relative humidity were calculated using the strong assumption that the daily dew point temperature was identical to the daily minimum temperature. This assumption was used in many studies and seems to be reasonable for most applications (Kimball et al., 1997; Thornton et al., 2000; Waichler and Wigmosta, 2003; Walter et al., 2005).

$$T_{dew}(t,z) = T_{min}(t,z) \tag{16}$$

The daily mean vapor pressure can be calculated from the estimated daily dew point temperature using the Tetens equation (Monteith and Unsworth, 2007). Figure 6 shows the average of the simulated and the observed dew point temperatures in relation to the elevation of the stations.

[FIGURE 6]




Figure 6 shows that for elevations from 400 to approximately 1250 m, the assumption that the daily dew point temperature matches daily minimum temperature is quite good on an annual average, with the 5th and 95th quantiles matching as well. However, this assumption shows lower accuracy at higher elevations, leading to considerable overestimation of the simulated mean annual dew point temperature and vapor pressure. This phenomenon can also be observed in Figure 7.

[FIGURE 7]

Figure 7 shows that the $R^2$ value between the simulation and observation of the vapor pressure tends to deteriorate when simulating higher elevations. Although the Figure 6 seems to show a constant bias at high elevations for the average annual values, Figure 7 shows that daily $P_{vap}$ has a different trend. At higher elevations, the occurence of the daily simulated vapor pressure matching the observation, therefore $T_{min}$ matching the daily $T_{dew}$, is much lower than at lower elevations. This means

that at high elevations the minimum daily temperature temperature is rarely so low as to reach the dew point, humidity being the limiting factor.

## 4   Results

The goal of this section is to compare the performance of the newly created parameterizations with the original Bristow transmissivity and Sicart emissivity equations combined. The parameter sets from the different radiation models are calibrated

on observed SW and LW data from 18 meteorological stations; the others are kept for validation purposes. These stations comprise a good sample of stations having different elevations and topographic morphologies.

The calibrations made in this section were done using a prefiltering method followed by the steepest descent calibration algorithm (?Perrin et al., 2001). This algorithm showed good results for parsimonious hydrological models, but the number of evaluations needed increases significantly with the number of parameters, although for this case study the computing time was

reasonable. The KGE' (Gupta et al., 2009; Kling et al., 2012) criterion was used to evaluate the performance of the calibration process (Eq. 17). This criterion takes into account the Pearson correlation coefficient (Eq. 18), the percentage bias (Eq.19) and the ratio of the coefficient of variation (Eq. 20).

$$KGE' = 1 - \sqrt{(r-1)^2 + (\omega-1)^2 + (\gamma-1)^2} \qquad (17)$$

$$r = Cov_{so}/(\hat{\sigma}\sigma) \qquad (18)$$

$$\omega = \hat{\mu}/\mu \qquad (19)$$

$$\gamma = \frac{\hat{\sigma}/\hat{\mu}}{\sigma/\mu} \qquad (20)$$

$\mu$ ($\hat{\mu}$) is the average of the population (simulation), $\sigma$ the standard deviation and $Cov_{so}$ the covariance between the observation and the simulation.

The results of this study are further analyzed using Friedman statistical test (Friedman, 1937) to determine the statistical

significance of the differences between the experiments tested.



All results presented in the following are validation results from stations that were not used for the calibration process. Using 18 stations for calibration, 90 SW stations and 16 LW stations remain for the validation set.

### 4.1 Comparison of the different parameterizations

In this section we compare the effect of the SW parameterizations presented in Table 2 on both SW and LW simulations.
We therefore chose to calibrate all the parameters of the different formulations together. The average between the KGE' performance on SW radiations and the KGE' performance on LW radiations is used as the evaluation criterion. Using the most complex shortwave parameterization proposed in this article (Eqs. 4, 8, 9) and the existing Sicart parameterization (Eqs. 14, 15, 16), nine parameters need to be calibrated ($\tau_{max}(z=0)$, $z_{ref.1}$, $\Delta T_{param}$, $\delta_{ref}$, $z_{ref.2}$, $\epsilon_1$, $\epsilon_2$, $RH_{ref}$, $\tau_{ref}$). Since there are two different $z_{ref}$ parameters in the $\tau_{max}(z)$ and the $\Delta T_{param}$ formulations, they were respectively called $z_{ref.1}$ and $z_{ref.2}$,
respectively.

The boxplots represented in Figure 8 show that the new formulations reproduce the SW and the LW radiations significantly better than the original Bristow formulation. The greatest improvement can be seen for high-elevation stations. The improvement comes from including both the topographic morphology in the parameterization and the absolute elevation ($B_1$). In contrast to section 3.1.2, it seems that using the absolute elevation only is better than using only the topographic morphology
indicator for simulating SW radiations. The median performance values for the new parameterization are 0.86 for SW and 0.88 for LW radiation simulation, while using the original Bristow formulation the simulations have median values of 0.83 for SW and 0.87 for LW.

[FIGURE 8]

To study how the parameterizations perform depending on elevation, the results were separated using three different elevation
categories (Fig. 8b, 8c, 8d). The results show better performance for SW with the new formulations for high elevations (median, 0.75 for $B_1$) compared to the reference Bristow formulation (median, 0.49). However, performance remained stable for low (median of 0.87) and for mid-elevations (median, 0.84) stations compared to the reference (median, 0.85 for both elevation classes).

Calibrating all the parameters simultaneously has the drawback of converging to parameter values that are unexpected. The
25 $\tau_{max}(z=0)$ parameter converged to a value of 1, meaning that the calibration tends to deactivate the $\tau_{max}(z)$ parameterization (Eq. 5). The $RH_{ref}$ parameter from the emissivity equation seems to be set for all calibrations at 0. This value deactivates the use of relative humidity in the model, showing that the simulated vapor pressure and relative humidity is noninformative for the model.

One possible way to avoid these problems is to use precalibrated parameter values to reduce the equifinality in the total
30 formulation.

Since some of the parameter values were estimated by regression in sections 3.1.1 and 3.1.2, they can be used as predetermined parameters in transmissivity parameterization, thus avoiding their calibration. Using the results of sections 3.1.1 and 3.1.2 gives values of 0.75 for $\tau_{max}(z=0)$, 2000 $m$ for $z_{ref.1}$, 0.544 for $m_{ref}$ and 8180 $m$ for $z_{ref.2}$. The only remaining parameter to calibrate for the transmissivity formulation is $\Delta T_{param}$, while all emissivity parameters remain to be calibrated.





The summary of these calibration methods is presented on Table 3 and the calibration results are available in Figure 9.

[TABLE 3]

[FIGURE 9]

Even when reducing the number of calibrated parameters, the value of the $RH_{ref}$ parameter of the Sicart emissivity equation
remains set to $0$. It seems to indicate that the improvement of the simulations of the transmissivity variable at high elevation has
little or no consequence on the emissivity simulations. It is therefore possible to assume that the estimation of the atmospheric
humidity (section 3.3) is the cause of the reduced performance at high-elevation stations. The results for the calibration after
regression showed a slight decrease in performance for SW (median KGE', 0.84) and a slight increase for LW radiations
(median KGE', 0.89) when reducing the number of calibrated parameters. Since the performance with the $B_{1,R}$ model remains
stable on average with the advantages of a substantially accelerated calibration, it seems to be the best choice for simulating
SW and LW radiations.

Figure 10 compares observed and simulated SW and LW time-series and Table 4 presents the RMSE, KGE and BIAS values
for the whole time-series available for these stations.

[FIGURE 10]

[TABLE 4]

Figure 10 shows better agreement of SW simulations with observations for Jungfraujoch station (JUN, $3580\ m$) using the
new parameterization compared to the original. This is confirmed by a much lower RMSE and bias value (Table 4). As seen
previously (Fig. 9), there are few benefits using the new transmissivity parameterization for LW simulations. For the Koppigen
station (KOP) located at a low elevation ($484\ m$), there is a slight degradation of the SW simulations, which are underevaluated
in the summer season. At the Samedan station (SAM, 1708 m) both parameterizations give very similar results, with little
performance degradation for the LW simulations.

Evaluating performance on measured SW and LW radiations shows the uncertainties that can occur on these variables.
However, it is difficult to assess how these uncertainties will impact the simulation of the thermal state of a surface. To evaluate
this uncertainty, we will force a reference surface with both observed and simulated radiation forcings.

## 4.2 Response of a reference surface to the simulated radiative forcings

The uncertainties of the SW and LW simulations were assessed and discussed in the previous section. However, when modeling
the energy balance processes, the response of the system to the uncertainties from the incoming radiations is rather complex.
Since the surface temperature is impacted by all the incoming energy fluxes, simulating surface temperature should highlight
the cascade of uncertainty coming from the radiative forcing used. The energy balance at a snow surface can be written as:





$$SW_\downarrow + LW_\downarrow + Pu_{prec} - SW_\uparrow(\alpha) - LW_\uparrow(T_s) - H(T_s) - E(T_s) - C(T_s) = 0 \tag{21}$$

$$SW_\uparrow(\alpha) = \alpha SW_\downarrow \tag{22}$$

$$C = \lambda \frac{\delta T}{\delta z}\Big|_{surface} \tag{23}$$

where $SW_\downarrow$ and $LW_\downarrow$ correspond to the incoming radiations that were parameterized in this paper, $SW_\uparrow$ and $LW_\uparrow$ cor-
respond to outgoing radiation depending on the snow albedo ($\alpha$) or surface temperature ($T_s$). The energy accounted for the
precipitation depends on the amount of precipitation $P$ ($kg.s^{-1}.m^{-2}$) and $u_{prec}$ ($J.kg^{-1}$). The conduction into the snowpack is
noted $C$; this flux depends on the thermal conductivity $\lambda$ ($W.m^{-1}.K^{-1}$) and the vertical temperature gradient in the snowpack
($K.m^{-1}$). The sensible ($H$) and latent ($E$) heat fluxes are also taken into account:

$$H(T_s) = \rho_a C_{p,a} \frac{T_s - T_a}{r_h} \tag{24}$$

$$E(T_s) = \rho_a \frac{q_s(T_s) - q_a}{r_v} \tag{25}$$

The aerodynamic resistance $r_h$ and $r_v$ ($s.m^{-1}$), the air density $\rho_a$ ($Kg.m^{-3}$), the heat capacity of air $C_{p,a}$ ($J.kg^{-1}.K^{-1}$),
the specific humidity at the surface $q_s$ ($Pa$) and the specific atmospheric humidity $q_a$ ($Pa$).

A simple method for evaluating uncertainties coming from the radiation inputs, inspired from the potential evapotranspiration
(PET) proposed by Allen et al. (1998), was developed. Since we intend to use the new parameterization mainly for snow
modeling, the impact of the simulation of SW and LW parameterizations will be assessed by calculating the response of a
reference surface. The characteristics of the reference snow surface are close to those of dry snow. This method needs the
following assumptions :

1. The reference surface optical properties are invariable in time, ie., the albedo is set at a constant value (0.7).

2. The reference surface is assumed to be a fully insulating material, meaning that no conduction is occurring in the bulk.

3. The aerodynamic resistance of the reference surface is also considered to be constant and has a low value ($340\ s.m^{-1}$),
given that snow has a lower atmospheric resistance than grassland.

4. The vapor pressure at the reference surface is considered to be equal to the saturation vapor pressure at the surface's
temperature.

5. The energy flux coming from precipitation is assumed to be low compared to the other energy fluxes and was considered
as negligible.





6. The temperature of the reference surface is allowed to exceed the value of $273.15\,K$, since it is an index of the energy supplied to the surface more than an actual snow surface temperature.

The reference surface and the forcings controlling the surface temperature are summarized in Figure 11.

[FIGURE 11]

For a better understanding of the processes, the model was computed at two moments each day, once at nighttime (at $T_{min}$) and once in the daytime (at $T_{max}$). We hypothesized that the time lag between $T_{max}$ and the maximum radiation is small. These assumptions allow computing the downward radiations using Eqs. 26 and 27:

$$nighttime \begin{cases} SW_{\downarrow min} = 0 \\ LW_{\downarrow min} = \epsilon \sigma T_{min}{}^4 \end{cases} \tag{26}$$

$$daytime \begin{cases} SW_{\downarrow max} = \tau R_{pot,max} \\ LW_{\downarrow max} = \epsilon \sigma T_{max}{}^4 \end{cases} \tag{27}$$

Since SW simulations are not used at nighttime, this method should help identify the uncertainties coming from the LW and SW simulations when simulating surface temperatures. The transmissivity ($\tau$) and emissivity ($\epsilon$) of the atmosphere can be calculated from observations or computed by the parameterizations developed above. The daily maximum potential SW radiation is computed for each day by integrating the analytical equation from Allen et al. (2006) at $30-min$ time-steps. Thus, the energy balance can be rewritten for nighttime and daytime using the following equations :

$$\begin{cases} LW_{\downarrow} - LW_{\uparrow}(T_{s,ref}^-) - H(T_{s,ref}^-) - E(T_{s,ref}^-) = 0;\ nighttime\ T_a = T_{min} \\ SW_{\downarrow} + LW_{\downarrow} - SW_{\uparrow} - LW_{\uparrow}(T_{s,ref}^+) - H(T_{s,ref}^+) - E(T_{s,ref}^+) = 0;\ daytime\ T_a = T_{max} \end{cases} \tag{28}$$

$T_{s,ref}^-$ and $T_{s,ref}^+$ ($K$) correspond to the daily minimum and maximum reference surface temperature. Two reference surface temperatures will be calculated and compared.

– $T_{s,ref}^+$ and $T_{s,ref}^-$ values coming from $\tau$ and $\epsilon$ values deducted using daily measured $SW_{\downarrow}$ and $LW_{\downarrow}$ radiations.

– $\widehat{T}_{s,ref}^+$ and $\widehat{T}_{s,ref}^-$ values coming from $\tau$ and $\epsilon$ values simulated by the parameterization method presented above.

The reference surface temperature was solved using the Newton-Raphson method. A KGE' performance criterion was computed in order to quantify the performance of the reference surface temperature using simulated SW and LW radiations. The reference surface temperature calculated using observed $\tau$ and $\epsilon$ was considered as the observation for computing the KGE' criterion. The performance on validation stations is available in Figure 12, and KGE, RMSE and BIAS values are given for Jungfraujoch, Koppigen and Samedan stations in Table 5.





[FIGURE 12]

[TABLE 5]

The $T_{s,ref}^{-}$ simulated at night (blue boxplots in Figure 12) show good performance for every model, with a slightly better performance for the $B_{1,R}$ model (median, 0.96). The performance is deteriorated at day time, with a much larger dispersion of the performance values than at the night time step (median, 0.93 for $B_{1,R}$). This seems to show that the SW radiation errors have a greater impact on the $T_{s,ref}$ than LW radiation simulations. This is confirmed by the fact that the performance of the simulation of $T_{s,ref}^{-}$ at night does not seem to deteriorate as much with elevation as the performance of LW simulations (Figure 9). The $B_{1,R}$ model also gives the best results for $T_{s,ref}^{+}$ simulations at day time. Unfortunately, the performance of the $B_{1,R}$ model for simulating the daily variation in $T_{s,ref}$ is slightly worse than the performance from other models for low-elevation areas. However, for higher elevations, the $B_{1,R}$ radiation model shows better performance than the others. For the three specific stations, Table 5 shows that performances is generally better using SW and LW radiations from $B_{1,R}$ for the Jungfraujoch station. For the Samedan and Koppigen stations, the performance using radiations from the $B_{1,R}$ is slightly lower than the performance using the original Bristow parameterization $B_{C}$.

## 5 Discussion

This new parameterization of atmospheric transmissivity decreased the errors of simulated SW radiations compared to the original Bristow formulation. However, even with an improved Bristow formulation, a number of uncertainties remain. The modeling of SW radiations defined in this paper depends on the daily temperature range, used as a proxy for cloud cover. However, the correlation between the daily temperature range and the cloud cover is not perfect since it the daily temperature range can be impacted by other parameters (winds, temperature inversions, etc.). In such cases, the model will simulate an incorrect value of transmissivity and directly impact the SW value simulated, because the daily temperature range gives misleading information. The parameterization of $\Delta T_{ref}$ seems to perform better for peak stations than for stations located in valleys. The $\Delta T_{ref}$ value does not go as high as it should in valley areas, probably affecting the transmissivity simulations for valley stations.

As the emissivity formulation used also depends on the cloud cover, simplified by the atmospheric transmissivity, the errors from the transmisivity simulations are transmitted to the LW simulations. Errors can also come from the estimation of the atmospheric vapor pressure, which is degrading when the elevation increases. Using the new transmissivity parameterization, the simulation of the LW radiations has improved very slightly, showing that transmissivity has less influence on the emissivity simulation than expected. It can be assumed that the lower performance of the LW simulations at high elevations compared to low elevations is mainly due to an overall overestimation of humidity in high-elevation areas (see Fig. 6).

For now, the model has not been tested on other data. The formulation was developed to be generic and not specific to a certain region or area. This means that if the model is used to simulate another region, the parameters should be calibrated using observations from the specific mountain range. Since the calibration method we propose takes multiple steps to evaluate parameter values, a regression analysis should also be done for any use of the model for another study site.



Even if the reference surface used in this paper uses many assumptions, it gives sufficient information about how improving the performance of SW and LW modeling impact the performance of simulation surface temperatures. Surprisingly, the KGE' performance values of reference surface temperatures are quite strong, showing that the retro-action of the snowpack from the radiative forcings smooths the uncertainties coming from the SW and LW simulations, with certain errors seeming to be compensated. It is also interesting to see that the differences of performance that can be seen between low and high elevations are also smoothed out when computing the reference surface temperature. Nevertheless, the validation set uses only 16 stations, because computing reference surface temperature performance criteria requires both LW and SW observations. The results taken from this section should be viewed with caution.

## 6 Conclusions

In hydrological studies, modeling snow and evapotranspiration often need energy forcing. As there are few measurements of the energy flux variables, these forcings are usually taken from atmospheric model simulations. Another approach for computing the incoming SW and LW radiations was tested in this study by parameterizing these fluxes using only the daily temperature range. The main objective of this study was to define a generic formulation of atmospheric transmissivity and emissivity, which can be adapted to specific regions by calibrating the parameters. This allows the use of an adaptive model that can be plugged into a hydrological model.

In this study we tried to improve an existing transmissivity parameterization in order to simulate SW and LW radiations with greater efficiency and genericity in regions with complex topography. Based on the Bristow and Campbell (1984) transmissivity parameterization, we attempted to increase its performance over 115 stations located in Switzerland. Adapting the original parameterization to take the elevation of the area considered into account gave better performance of SW simulations for high-elevation stations without excessively degrading low- and mid-elevation stations. The use of topographic morphology and elevation variables seems to give more information to the model than using monthly mean values as done in the Bristow's parameterization. This formulation using topographic information was tested and validated on 108 stations in Switzerland, with 18 stations used for calibrating the radiation model and the rest used for validating the SW (90 stations) or LW (16 stations) performance.

The LW radiations were also calculated using the Sicart et al. (2006) emissivity parameterization, which uses transmissivity as an input. The results showed that even with improved transmissivity simulations, the LW simulations remained identical. A better transmissivity simulation for high elevations is not sufficient to significantly improve the LW simulations. A preliminary study to verify the validity of the assumption that the daily dew point temperature is close to the daily minimal air temperature showed good humidity estimates at low elevations. However, for higher elevations, this assumption gives more errors, leading to more uncertainties in the LW simulations. These results show that LW radiations at high elevations could probably be much improved by improving the humidity simulation or by using humidity measurements.

The formulations were also validated by computing reference surface temperatures. These temperatures cannot be compared to actual snow surface temperatures due to strong assumptions as a constant albedo and zero conduction into snow. However,




this makes possible to take into account both SW and LW incoming radiations, with part of the retro-actions that take place in the energy balance. These simulations gave good performance with a slight improvement using the new parameterization over the Bristow's formula. This experiment also showed that errors in LW simulations have fewer impacts on the $T_{s,ref}$ simulation than SW simulations.

The main drawback of this new formulation is that a full calibration makes several parameters converge to values that are not physically consistent. This shows that the formulation is overparameterized. A solution to decrease this overparameterization is to determine the value of some parameters before calibration using regression coefficients from intermediate observations. This intermediary step before the calibration process helped to find more consistent parameters with only a slight decrease in performance compared to a fully-calibrated parameterization. The consistency of the parameter set was preferred over the

achievement of maximum performance.

A further step in the radiation model is to decompose the SW and LW radiation at a fine time step and to spatialize it on a DEM grid. The main difficulty spatializing the radiation resides in the fact that the daily temperature ranges are hard to spatialize. This radiation model will further be coupled to a new snow model being developed at UPMC. This is part of a project that has the objective of creating a new conceptual snow model based on a simplified energy balance. Although these

radiations have been simulated for mountainous snowy areas, they could also be tested as evapotranspiration model forcings.

*Competing interests.*   The authors declare that they have no conflict of interest

*Acknowledgements.*   The authors would like to acknowledge MeteoSwiss for providing the meteorological data used in this study. We would also like to thank Charles Obled and Remy Garcon for the thoughtful discussions on this particular subject.





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





**Table 1.** Summary of the different $\Delta T_{ref}$ formulations used in this study.

| Model | Abbreviation | $\Delta T_{ref}$ | Reference |
|---|---|---|---|
| Bristow $\Delta T_{ref}$ | $dT$ | $1/(0.036\, exp(-0.154\overline{\Delta T_m}))$ | Bristow and Campbell (1984) |
| Bristow simplified $\Delta T_{ref}$ | $dT_s$ | $k\overline{\Delta T_a}(M)$ | This article Eq. 6 |
| New $\Delta T_{ref}$ parameterization | $dT_1$ | $\Delta T_{param}\, exp\left(\frac{\overline{\delta}}{\delta_{ref}} - \frac{z}{z_{ref}}\right)$ | This article Eq. 8 |
| $\Delta T_{ref}$ topography | $dT_2$ | $\Delta T_{param}\, exp\left(\frac{\overline{\delta}}{\delta_{ref}}\right)$ | This article Eq. 11 |
| $\Delta T_{ref}$ elevation | $dT_3$ | $\Delta T_{param}\, exp\left(-\frac{z}{z_{ref}}\right)$ | This article Eq. 10 |





**Table 2.** Summary of the different transmissivity models used in this study, with the different $\Delta T_{ref}$ formulations used in this study.

| Model | Abbreviation | $\tau_{max}$ | $\Delta T_{ref}$ |
|---|---|---|---|
| Bristow model | $B$ | $A$ | $dT$, Bristow and Campbell (1984) |
| Bristow simplified model | $B_s$ | $\tau_{max}(z)$, Eq. 5 | $dT_s$, Eq. 6 |
| New transmissivity parameterization | $B_1$ | $\tau_{max}(z)$, Eq. 5 | $dT_1$, Eq. 8, 9 |
| Transmissivity topography parameterization | $B_2$ | $\tau_{max}(z)$, Eq. 5 | $dT_2$, Eq. 11 |
| Transmissivity elevation parameterization | $B_3$ | $\tau_{max}(z)$, Eq. 5 | $dT_3$, Eq. 10 |



**Table 3.** Summary of the different calibration methods used. The "C" flag coresponds to a parameter that has to be calibrated. Each parameter from the combined calibration has to be calibrated.

| Calibration type | Abbreviation | $\tau_{max,z=0}$ | $z_{ref.1}$ | $m_{ref}$ | $z_{ref.2}$ | $\Delta T_{param}$ | $\epsilon_1, \epsilon_2, RH_{ref}, \tau_{ref}$ |
|---|---|---|---|---|---|---|---|
| Combined calibration | $B_{1,C}$ | C | C | C | C | C | C |
| Calibration after regression | $B_{1,R}$ | 0.75 | 2000 | 0.544 | 8180 | C | C |





**Table 4.** KGE, RMSE and BIAS performance of the new $B_1$ model calibrated with the regression method and the Bristow reference model $B_C$.

| | | $B_C$ | | | $B_{1,R}$ | | |
| --- | --- | --- | --- | --- | --- | --- | --- |
| | | KGE | RMSE | BIAS (%) | KGE | RMSE | BIAS |
| JUN | SW | 0.485 | 96.6 | -39.7 | 0.811 | 58.5 | -7.33 |
| | LW | 0.387 | 42.8 | 1.86 | 0.388 | 42.0 | -2.92 |
| KOP | SW | 0.841 | 39.7 | 12.6 | 0.850 | 35.6 | 3.36 |
| | LW | 0.863 | 23.1 | -5.08 | 0.907 | 18.8 | -3.29 |
| SAM | SW | 0.846 | 45.88 | 8.36 | 0.843 | 44.5 | 9.15 |
| | LW | 0.874 | 30.91 | -9.31 | 0.879 | 31.9 | -9.98 |



**Table 5.** KGE, RMSE and PBIAS performance of the simulated daytime $T_{s,ref}^{+}$, nighttime $T_{s,ref}^{-}$ and daily $\Delta T_{s,ref}$ range using the simulated SW and LW radiations based on the $B_{1,R}$ model and the Bristow reference model $B_C$.

|  |  | $B_C$ | | | $B_{1,R}$ | | |
|---|---|---|---|---|---|---|---|
|  |  | KGE | RMSE | BIAS (%) | KGE | RMSE | BIAS (%) |
| JUN | $T_{s,ref}^{-}$ | 0.85 | 4.5 | 0.16 | 0.85 | 4.4 | -0.23 |
|  | $T_{s,ref}^{+}$ | 0.74 | 6.2 | -1.90 | 0.83 | 3.6 | -0.92 |
|  | $\Delta T_{s,ref}$ | 0.39 | 7.5 | -35.00 | 0.58 | 5.1 | -12.0 |
| KOP | $T_{s,ref}^{-}$ | 0.92 | 1.7 | -0.40 | 0.96 | 1.4 | -0.25 |
|  | $T_{s,ref}^{+}$ | 0.85 | 1.6 | -0.31 | 0.93 | 1.5 | -0.32 |
|  | $\Delta T_{s,ref}$ | 0.90 | 1.7 | 2.30 | 0.87 | 1.7 | -2.1 |
| SAM | $T_{s,ref}^{-}$ | 0.95 | 2.7 | -0.79 | 0.95 | 2.7 | -0.84 |
|  | $T_{s,ref}^{+}$ | 0.78 | 2.8 | -0.76 | 0.82 | 2.7 | -0.70 |
|  | $\Delta T_{s,ref}$ | 0.85 | 2.2 | -0.2 | 0.86 | 2.1 | 0.84 |



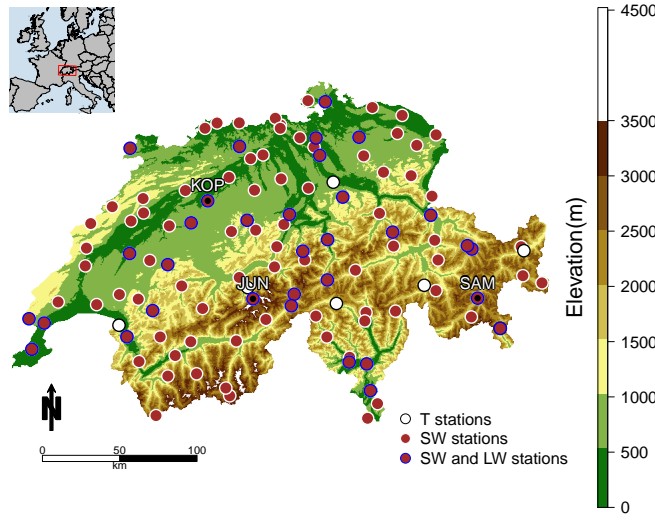

**Figure 1.** Location of the 115 meteorological stations used for this study. The locations of Jungfraujoch (JUN), Koppigen (KOP) and Samedan (SAM) used in the Result section are given.

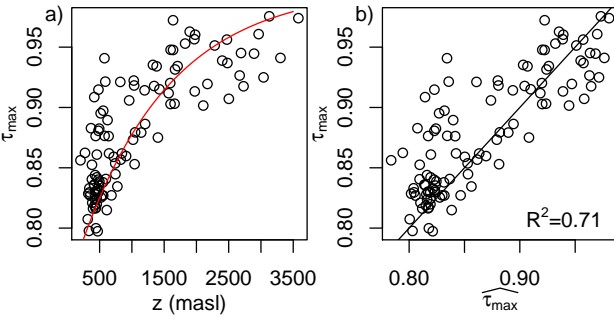

**Figure 2.** Parameterization of $\tau_{max}(z)$. a) Changes in the observed maximum transmissivity with elevation and its simulation from Eq. 5 (red line). b) Scatterplot of observed and simulated maximum transmissivity.





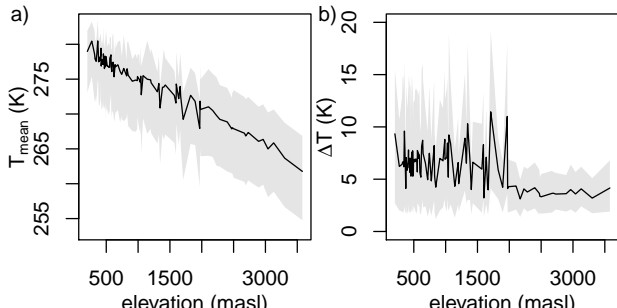

**Figure 3.** Mean annual temperature (a) and mean daily temperature range (b) versus elevation. Shaded area represents the $5^{th}$ and $95^{th}$ percentiles of these two variables.





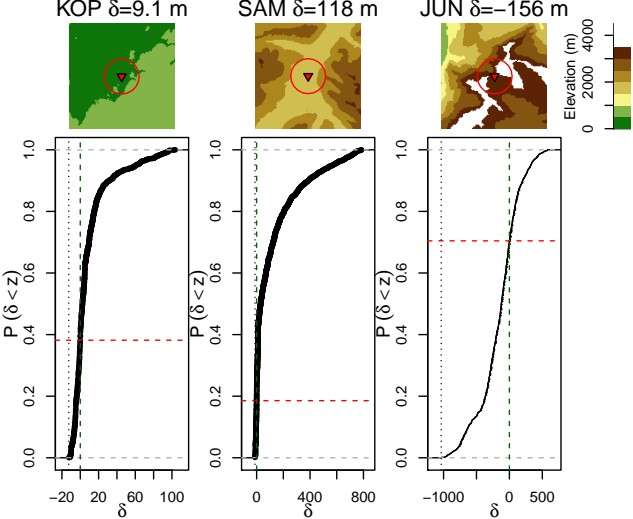

**Figure 4.** Distribution of $\delta$ values within 2000 m around three meteorological stations: Koppigen (KOP, 484 m), Samedan (SAM, 1708 m) and Jungfraujoch (JUN, 3580 m). The DEM associated for each station is provided at the top of the distributions, the circle indicating the 2000 $m$ buffer area. The average of the distribution is noted $\overline{\overline{\delta}}$. The horizontal red dashed line corresponds to the probability of non-exceedance of the station's elevation.





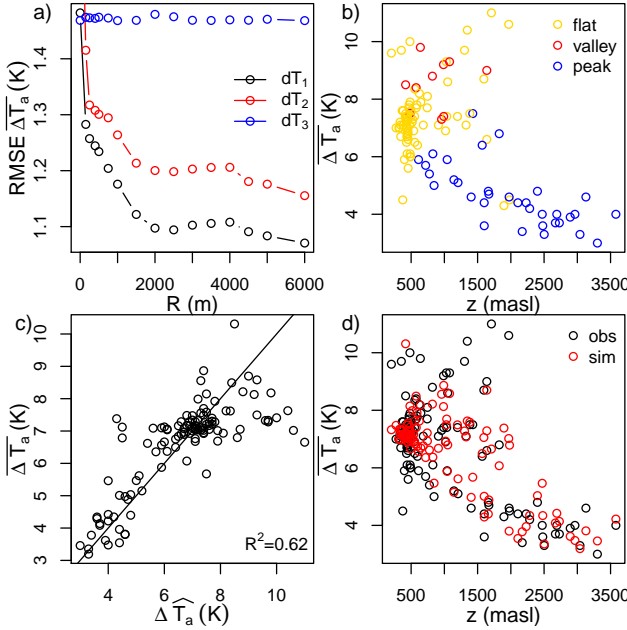

**Figure 5.** Performance analysis for reproducing the mean mean temperature range ($\overline{\Delta T_a}$) value for each meteorological station. a) RMSE performance between observed and simulated $\overline{\Delta T_a}$ versus the value of the radius buffer used for computing the $\overline{\overline{\delta}}$ variable. b) Discretization of the stations into three different topographic morphologies by the model; the type of morphologies were attributed using a minimum of 10% changes compared to the reference $\overline{\Delta T_a}(z = 0, \overline{\overline{\delta}} = 0)$ value. c) Scatterplot of the observed mean $\overline{\Delta T_a}$ compared to its simulation $\widehat{\Delta T_a}$ and d) comparison of observed and simulated $\overline{\Delta T_a}$ values for all meteorological stations.





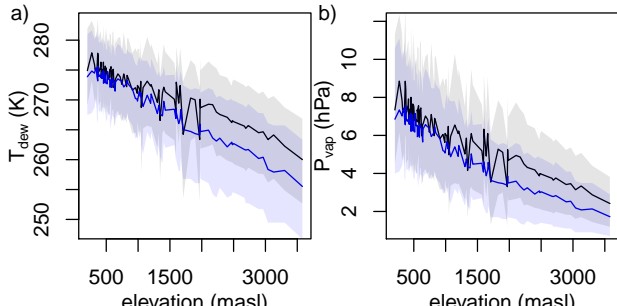

**Figure 6.** Analysis of humidity versus elevation. a) Comparison of the observed mean annual dew temperature (blue) and the observed mean annual daily minimum temperature (black) and b) comparison of the observed mean annual vapor pressure (blue) and the simulated vapor pressure using Eq. 16 (black). The shaded areas show the variability of the daily values ($5^{th}$ and $95^{th}$ quantiles).





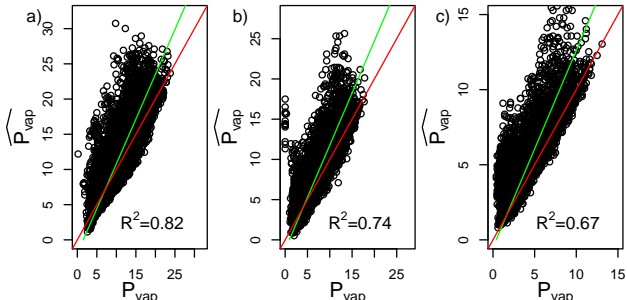

**Figure 7.** Scatterplots of the daily observed vapor pressure ($P_{vap}$, $hPa$) and the daily simulated vapor pressure ($\widehat{P}_{vap}$, $hPa$), a) for low-elevation stations (0 to 1000 m), b) for mid-elevation stations (1000 to 2000 m), c) for high-elevation stations (above 2000 m).





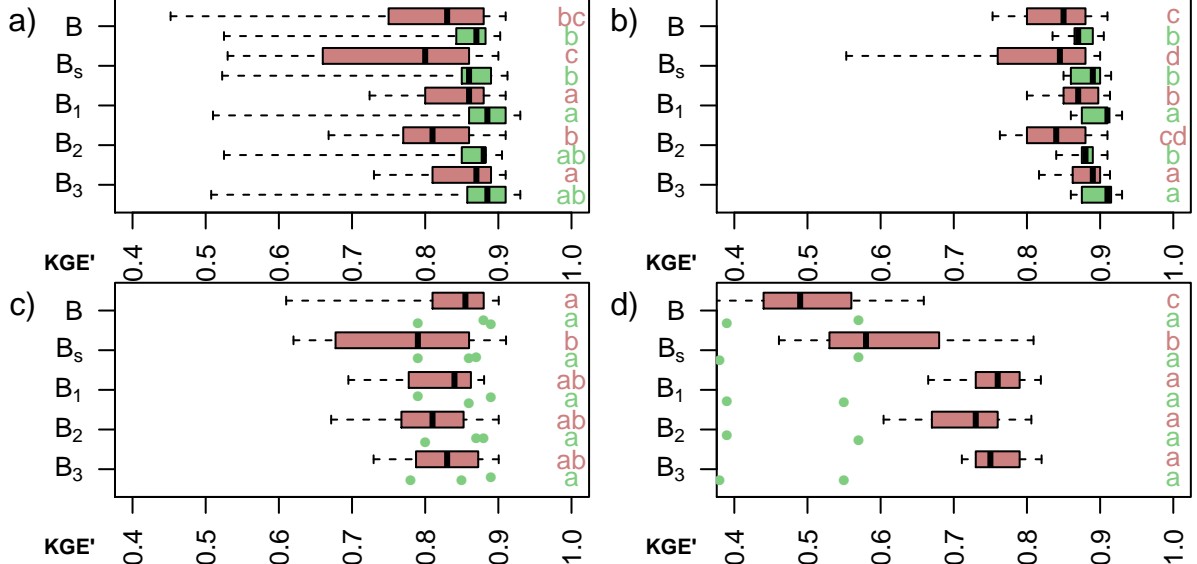

**Figure 8.** Performance of validation stations using different SW parameterizations presented in Table 2. a) Overall performance for all stations; b) performance for low-elevation stations; c) performance for mid-elevation stations; d) performance for high-elevation stations. All parameters were calibrated at the same time. Red boxplots indicate the performance of SW simulations and green boxplots indicate the performances of LW simulations. The letters at the right of the boxplots indicate the results of the Friedman statistical test. The "a" indicates the best model according to the statistical model and two identical letters indicate no significant differences between the models.





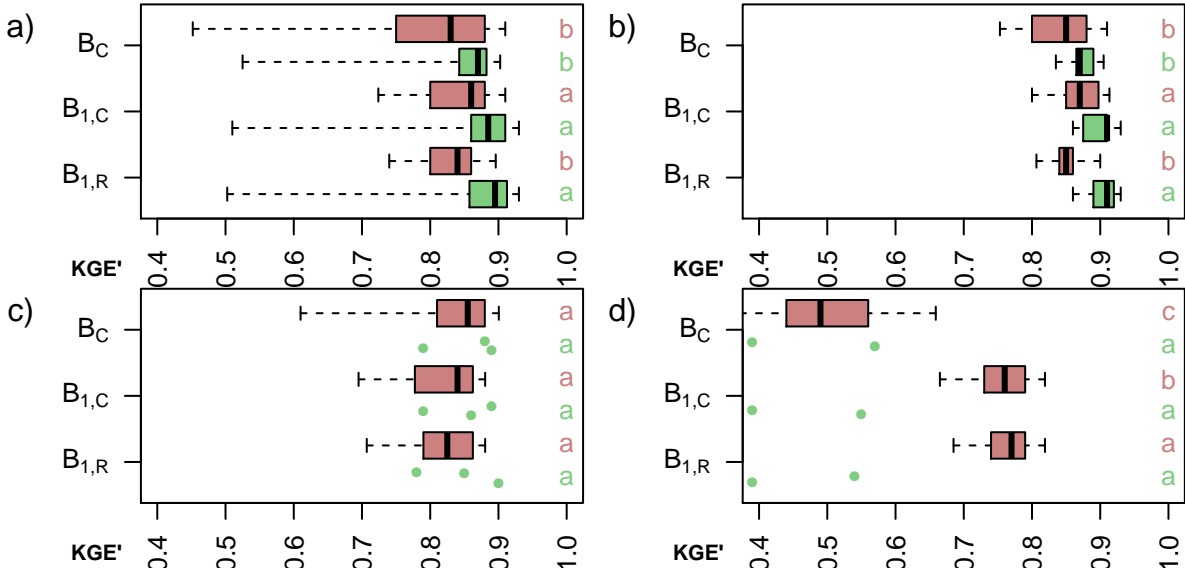

**Figure 9.** Performance of validation stations using different calibration methods. The new parameterization $B_1$ is compared to the reference Bristow formulation ($B$). a) Overall performance for all stations; b) performance for low-elevation stations; c) performance for mid-elevation stations; d) performance for high-elevation stations. Red boxplots indicate the performance of SW simulations and green boxplots indicate the performances of LW simulations. Regression parameters were only computed for the $B_1$ parameterization.





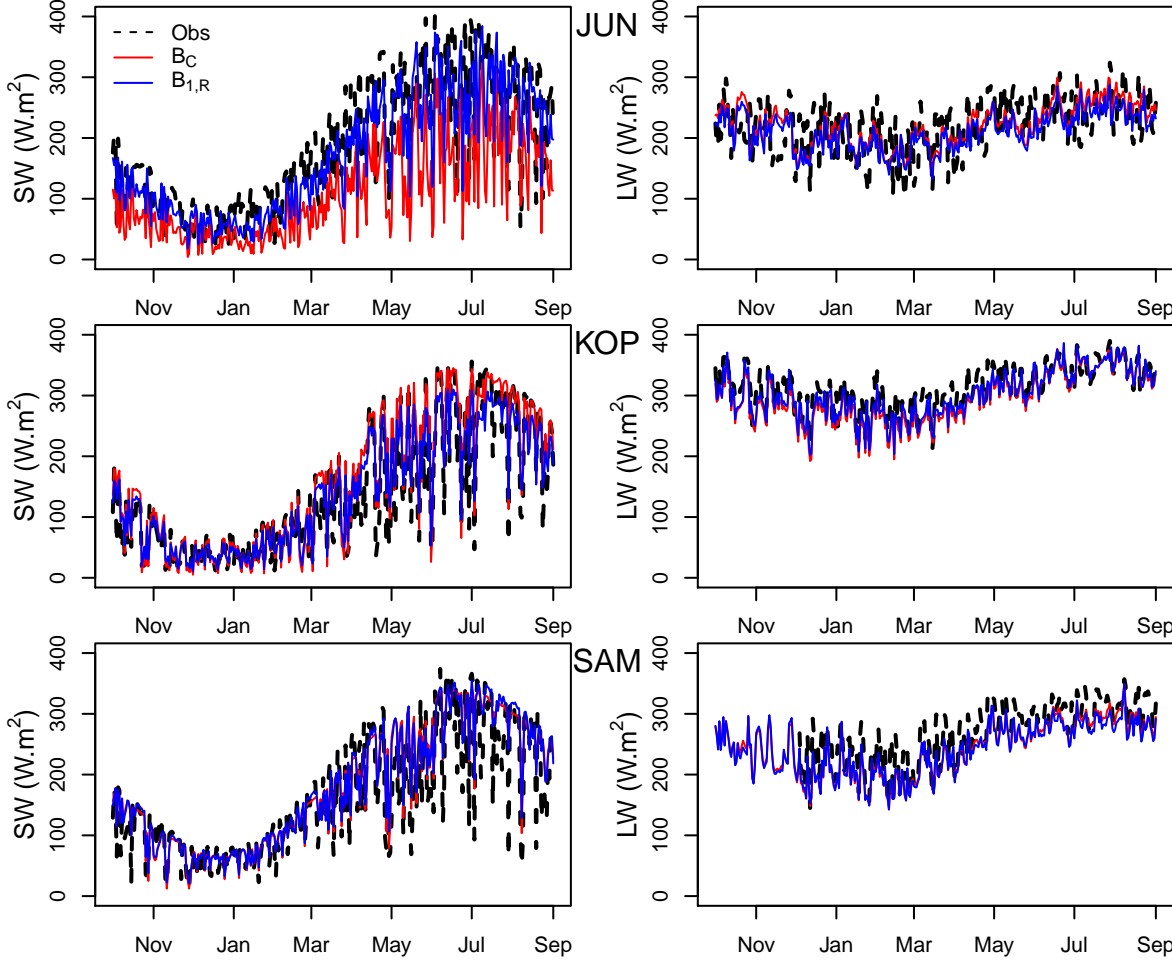

**Figure 10.** Comparison of observed and simulated SW and LW time-series for three meteorological stations (Jungfraujoch (JUN), Koppigen (KOP) and Samedan (SAM)) using the original Bristow and the newly developed parameterizations. The time series presented are from October 2012 to July 2013.



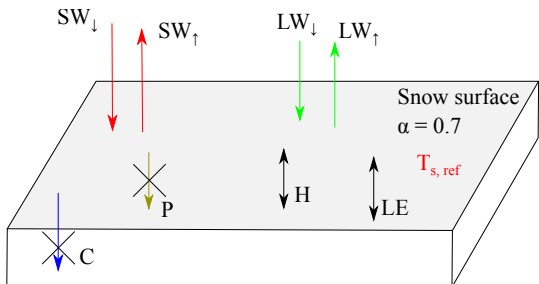

**Figure 11.** Representation of the experiment designed to compute the reference surface temperature index (Eq. 28). Crossed processes are neglected. Values for heat fluxes are considered positive when their direction goes from the snow to the atmosphere.

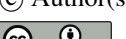


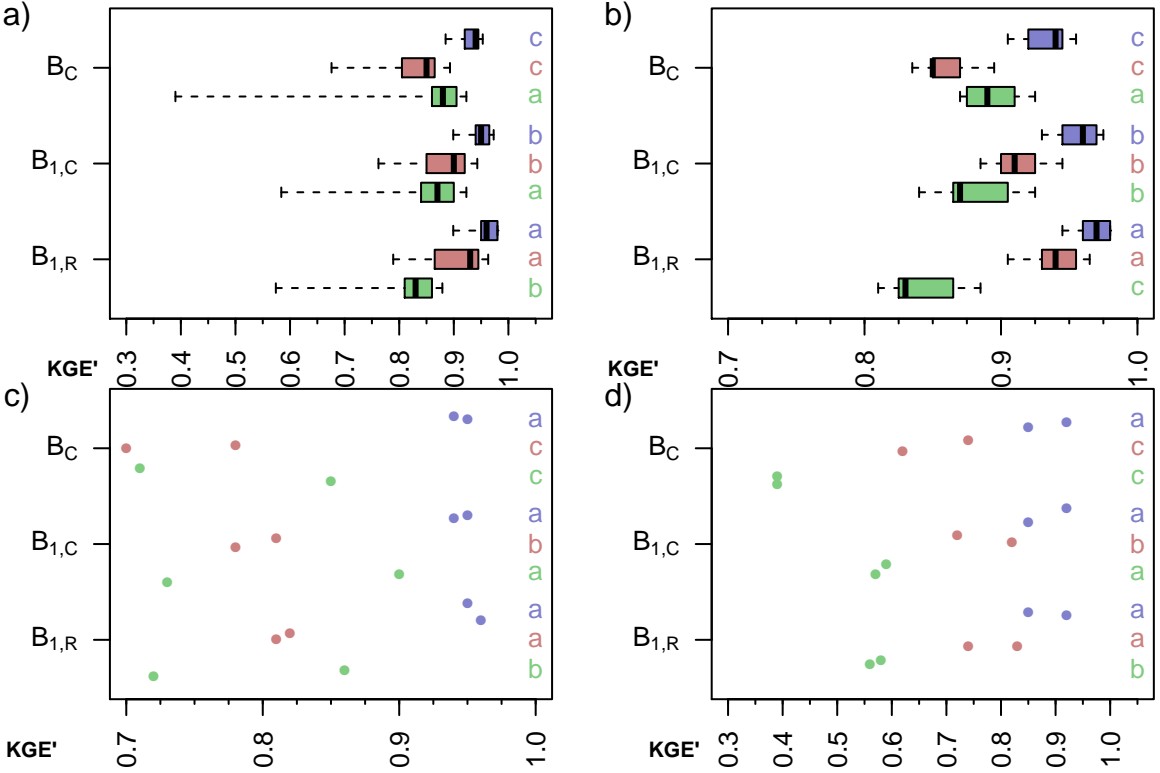

**Figure 12.** Performance of validation stations for simulating reference surface temperature. The KGE' performance was computed versus $T_{s,ref}$ values using observed forcings. a) Overall performance for all stations; b) performance for low-elevation stations; c) performance for mid-elevation stations; d) performance for high-elevation stations. Blue boxplots represent the performance computed on daily minimum $T_{s,ref}^{-}$, red boxplots on daily maximum $T_{s,ref}^{+}$ and green boxplots on daily surface temperature $\Delta T_{s,ref}$ range.