# Peer review of "How to simulate radiative inputs in complex topographic areas, an analysis on 115 Swiss Alps weather stations"

_Hydrology and Earth System Sciences, 2017_

## Referee Comment (RC1) · Anonymous Referee #1 · 27 Oct 2017

General comments:

The paper deals with, in my view, an important topic; "the introduction of more physically based (snow) models while acknowledging a lack of forcing data". I believe this is the way forward to make progress both for PUB and for predicting hydrology under a changed climate. As I read the paper, the study tries to improve the transmissivity algorithm of Bristow and Campbell by taking into account elevation and topography. This would obviously improve the SW estimation and, through the Sicart formula for emissivity, the improved transmissivity would improve the estimation of atmospheric LW. The chosen method involves a lot of parameters that need to be calibrated (i.e.

[Figure]

needs a lot of location specific (?) data, which we are trying to avoid) and even if the method gave improved SW- estimates for high altitudes, the results for lower elevations and for LW were not as was hoped for. In its current state, this is not a method I would implement or can recommend. I do think, however, that using all relevant, easily obtainable information (for example elevation and topography) to improve the algorithms for more physically based models, such as energy balance (EB) modelling, is a good idea. I believe the paper needs major revision, and possibly some revisions in the method as well before it can be considered for publication. The following points need to be addressed:

1) I think, but I am not sure as it is not clearly stated, that the temporal scale is daily. For snow models, the potential gain in using EB models is for finer temporal scales. At daily time scales well calibrated (against snow data) degree-day models does the job quite nicely. I believe this was the conclusions of Anderson (1976-77). This point needs to be addressed quite early and revisited in the discussion.

2)You introduce a lot of parameters to be calibrated and state yourself that the model is overparameterised. If some physically based reason (model without calibration parameters) for why, for example $\tau$_max increases with altitude could be introduced, then the model will have less freedom and be easier to diagnose (why doesn't it work). The dependence on calibration is also a problem for using the models for other temporal resolutions (where to find the data?).

3)I think the paper has a serious problem with notation. It is very confusing to have the letter "T" surrounded by some many super- and subscripts. I quickly lost track over what was what and this is especially difficult in 4.2. Notations like ãĂŰ△TãĂŮ_param and T_(s,ref)ˆ+ are not very helpful. Have the units in [] instead of (). Explain the variables directly after an equation (see deltaTref at page 4 L20 and Eq. 9). In Eq. 14 you have three different epsilons, are all emmisivities?

4)I kept wondering why you have 4.2. It comes as an extra exercise at the end of the

paper and I did not think it helped to clarify matters.

5)Can you discuss your method in relation to what has been done internationally in the field?

Specific comments:

Title: "How to simulate.." appears a bit too confident. How about the somewhat more humble "Empirical modelling of radiative . . .

Abstract„ L3 : change scarce for coarse? and ..fluxes in areas of complex topography Introduce the temporal resolution in the abstract.

P1.L13: what is retro-action, do you perhaps mean feedback??

P2.L33: spell out UEB and this sentence need reformulation, . . .which signal?

P3.L1 You say that errors on LW have a great impact on SWE(P2.L34), and conclude that it doesn't?

P3.L14 Here is the temporal resolution first introduced. Should be much earlier. I think you can have a more thorough outline of the study in the introduction.

P4.L1 with Rpot being the. . .

P4.L13 space dependent

P4.L20: reformulate.. a parameterization for the reference parameter for the daily temperature range.

P5.L11. start the paragraph with We want to ..

P5.L12 Fig3b?

P5.L20 . . .parameter is proportional to the mean..

P5.L22 Figure 3a?

P5.L23 it appears quite linear to me..

P5.L26 ..range, and mountain..

P5.L30 the differences in elevation is denoted $\delta$

P6.L4 point instead of area? And the average elevation difference noted.. How did you test?

P6.L7 If $\delta$ Ì Ě... and so on for the next lines

P6.L19 which is the average

P6.L21 We do not know that the range depends on...

P7.L30 reformulate sentence

P8.L18.. humidity is needed.

P8.L24 What happened to Tetens equation?

P9.L8 Pvap not defined

P9.L10 ..point, as humidity is the ..

P9.L15. Be specific about the stations, not "others" and "these"..

P9.L27 mu(mu)???

P12.L1 u_prec??

P12. Eq. 24,25 no wind speed?

P12.L25.. is assumed to be negligible compared to other energy fluxes

P15.L3 retro-action again

P15.L17 generality

Figure 6. Eq. 16 does not yield vapor pressure

Figure 8 what are the circles. What does the letter indicate? There are no two identical letters. Do you really need to compare so many models? We lose track of which is which with the uninformative abbreviations.

Figure 12. Lots of unexplained dots
* * *

---

## Referee Comment (RC2) · Anonymous Referee #2 · 5 Nov 2017

The authors used data from over 100 weather stations with different aspect and altitude to develop more accurate long and short-wave radiation estimates from readily-available data such as temperature and geographic features. While I believe this to be a worthwhile task, the manuscript as written is hard to understand, and unclear in its major findings. Consequently, I recommend revisions to the paper before it can be published. First – it was hard to follow all your descriptions of the various parameters used. I'd suggest putting some of the detail in an appendix section, and simply providing the parameter formulations you ended up using in the main documentation. Also – I'd like to see clearer comparisons between the new model results and more established methodologies for estimating LW and SW radiation. The use of the KGE

criterion is unclear – why not use Pearson correlations directly? If this criterion is indeed superior, please provide explanation of acceptable ranges and max/min values. In addition, it would be useful to see how the provided formulations compare to simply using the nearest weather-station values for LW and SW radiation, or reanalysis datasets – i.e provide some more broad comparisons for your methodology. In the end, it didn't appear that your calibrated models performed much better than the original models. For your conclusion - discuss conditions under which using these more complex formulations would be worthwhile.

Specific comments: P8L14: Semicolon needed P9L18: extra "?" P9L23: Check this equation, it doesn't seem correct in the form written, as it equates three measures of effectiveness similarly in the equation, even though one should be maximized, the others minimized. Table 4: What are JUN, KOP, and SAM? Please define the acronyms and their significance in the table caption (i.e. do these represent high altitude, mid, and low altitude stations?).

---

## Author Comment (AC1) · 14 Dec 2017

Response to Anonymous Referee #1

We would like to thank the reviewer for his/her insightful comments and suggestions. We are sure that taking these suggestions into account will significantly improve the clarity and the scientific interest of this paper. We try to address them below in a point-by-point format. The original review is presented in black and the response is in red.

General comments:

The paper deals with, in my view, an important topic; "the introduction of more physically based (snow) models while acknowledging a lack of forcing data". I believe this is the way forward to make progress both for PUB and for predicting hydrology under a changed climate. As I read the paper, the study tries to improve the transmissivity algorithm of Bristow and Campbell by taking into account elevation and topography. This would obviously improve the SW estimation and, through the Sicart formula for emissivity, the improved transmissivity would improve the estimation of atmospheric LW. The chosen method involves a lot of parameters that need to be calibrated (i.e. needs a lot of location specific (?) data, which we are trying to avoid) and even if the method gave improved SW- estimates for high altitudes, the results for lower elevations and for LW were not as was hoped for.

Response: The original Bristow and Campbell and Sicart formulations were developed for specific areas and the respective authors proposed parameter values for these formulations. We believe that the formulations of emissivity and transmissivity provided can be generalized at different mountain ranges by recalibrating their parameters, provided that data are available. We used a set of 18 stations for calibration, which seemed to be a moderate amount of stations for representing the Alpine region. We will try to emphasize better the added value (generality) of the method. Another potential advantage is that this methodology can also be applied in the absence of radiation data: when using these formulations together with snow and/or hydrological models, the parameters from the SW and LW formulations could also be calibrated conjointly with the parameters of the snow and/or hydrological model on snow or discharge measurements. Finally, satellite temperature data could also be used to calibrate the formulations. We believe that the multiplicity of potential applications of this work make this methodology worth being developed and published.

In its current state, this is not a method I would implement or can recommend. I do think, however, that using all relevant, easily obtainable information (for example elevation and topography) to improve the algorithms for more physically based models, such as energy balance (EB) modelling, is a good idea. I believe the paper needs major revision, and possibly some revisions in the method as well before it can be considered for publication. The following points need to be addressed:

1) I think, but I am not sure as it is not clearly stated, that the temporal scale is daily. For snow models, the potential gain in using EB models is for finer temporal scales. At daily time scales well calibrated (against snow data) degree-day models does the job quite nicely. I believe this was the conclusions of Anderson (1976-77). This point needs to be addressed quite early and revisited in the discussion.

Response: The simulations provided by these formulations are indeed at daily time steps. Many transmissivity and emissivity parameterizations are using this time step. For developing and

comparing a new transmissivity function to the original Bristow formulation, we chose to stay at this time step. As pointed out by the reviewer, at daily time step, degree-day snow models give good results in terms of snowmelt. However, we believe that this work can be the first part of the development of a sub-daily snow model, which would benefit from a sub-daily SW and LW incoming simulation. This should allow better simulating of snowmelt processes compared to degree-day models, as it would improve the space-time variability of the melt. This will need to disaggregate daily temperature values and calculate the potential solar radiation at sub-daily time steps. The simulated transmissivity and emissivity should be interpolated between days or be considered constant in order to generate sub-daily SW and LW values. A first glimpse of this disaggregation is showed on section 4.2, where we calculate the minimal and maximal daily reference surface temperature. Nevertheless, we agree that it is needed to add a discussion about the time steps used in this article and the importance they have on snow modelling. We will discuss it further in a revised version of the manuscript and better mention the model time step earlier in the manuscript.

2) You introduce a lot of parameters to be calibrated and state yourself that the model is overparameterised.  If some physically based reason (model without calibration parameters) for why, for example τ_max increases with altitude could be introduced, then the model will have less freedom and be easier to diagnose (why doesn't it work). The dependence on calibration is also a problem for using the models for other temporal resolutions (where to find the data?).

Response: We agree that calibrating all parameters can be cumbersome due to overparameterization. In order to avoid this issue, we proposed an alternative method in the manuscript that was analyzed in the result section 4.1. For this method, which was introduced in the parameterization development, only few parameters are calibrated. In addition, we use regressions to determine meta-parameters. We consider for instance that the τ_max formulation is independent from the rest and its parameter can be fixed on their own by looking only on the τ_max observed values of each station. The same has been done for the parameters driving the topography indices.

In order to improve the clarity of the paper and give more visibility to this alternative method, its description will be improved and the experimentation plan will be specified in a new subsection (including first parameterization with full calibration and then the use of regression, together with a couple of other related items). In the results section, the regression method results will be separated in a dedicated subsection (i.e. subsection 4.1 will be reorganized).

Concerning a physically-based τ_max formulation, Thornton and Running developed a similar equation. It depends on the atmospheric pressure, the optical air mass and the vapor pressure. This formulation, which uses more physical parameters (atmospheric pressure and optical air mass), depends only on one parameter.  If judged necessary, using part of this formulation could be tested in order to replace or to be compared to our τ_max formulation.

3) I think the paper has a serious problem with notation.  It is very confusing to have the letter "T" surrounded by some many super- and subscripts. I quickly lost track over what was what and this is especially difficult in 4.2. Notations like ã̌Ă˝UΔTã̌Å˚U_param and T_(s,ref)ˆ+ are not very helpful. Have the units in [] instead of ().   Explain the variables directly after an equation (see deltaTref at page 4 L20 and Eq. 9). In Eq. 14 you have three different epsilons, are all emmisivities?

Response: The authors acknowledge that the parameter naming could be improved as indicated as many of them have the subscript "ref", which is also used by the reference surface temperature. We chose to use subscripts and hyperscripts in order to better understand the meaning of each of the parameters. For instance $\Delta T_{param}$ gives the following information that this parameter manage the evolution of the transmissivity for values of daily temperature range $\Delta T$ and has the same units as $\Delta T$.

We intend to give to the calibrated parameters the subscript "p" (for instance $z_p$ instead of $z_{ref}$). This should also avoid any misunderstanding with the reference surface temperature $T_{s,ref}$. In addition, the intermediate variable will be using the "c" subscript (meaning "characteristic value") instead of "ref".

4) I kept wondering why you have 4.2. It comes as an extra exercise at the end of the paper and I did not think it helped to clarify matters.

Response: The 4.2 section is a manner to understand how errors coming from the SW and LW simulations impact the surface temperature. All in all, modelers are not directly interested on the SW and LW simulations, but much more about the simulation of snowmelt. In the 4.2 section we provide a simple benchmark system in order to understand the feedbacks of the whole snow surface system to the simulated incoming radiations. As a consequence, we would like to keep this section. In order to improve the quality of the manuscript, a method section will be added, in which the parameterization development and the description of the reference surface will be moved in a revised version of the manuscript. Only the results of the simulation of the reference surface temperature will be kept in the result section.

5) Can you discuss your method in relation to what has been done internationally in the field?

Response: The authors acknowledge that little comparison has been made with other formulations in this paper in order to limit the size of it. Nevertheless, a discussion of some previous works will be made in the discussion section. If judged necessary, a comparison of different existing parameterizations could be done in the appendix section, but we think that it would increase the complexity of the paper.

Specific comments:

Title: "How to simulate.." appears a bit too confident. How about the somewhat more humble "Empirical modelling of radiative...

Response: We agree with your comment, your proposition is more suitable than the title we proposed.

Abstract,, L3 : change scarce for coarse? and ..fluxes in areas of complex topography Introduce the temporal resolution in the abstract.

Response: It will be modified

P1.L13: what is retro-action, do you perhaps mean feedback??

Response: Yes, it will be modified

P2.L33: spell out UEB and this sentence need reformulation,...which signal?

Response: UEB means Utah Energy Balance snow model. We propose the following modification to the sentence "Lapo et al. (2015) used the UEB snow model to quantify the impact of inaccuracies in incoming radiation simulations. They showed that, for LW simulations, bias errors have a greater impact than standard deviation errors on the simulation of snow water equivalent (SWE) and surface temperature."

P3.L1 You say that errors on LW have a great impact on SWE (P2.L34), and conclude that it doesn't?

Response: P2L34 we say that LW simulation bias errors have more impacts than LW standard deviation errors for SWE simulations. In this sentence, we wanted to say that one should be careful in the performance criteria used to validate SW and LW results as improving simulated LW standard deviation should have a lower impact on SWE simulation than reducing the bias. We propose to replace this sentence by the following: "Thus, improving SW and LW formulations does not automatically translate into improving the SWE simulation, as it depends on the performance criteria chosen (in this case a criteria with more weight on standard deviation may increase bias). (Lapo et al., 2015) showed that the surface temperature is a good indicator of snow model performance, as a validation on SWE only is unable to indicate energy fluxes errors."

P3.L14 Here is the temporal resolution first introduced. Should be much earlier. I think you can have a more thorough outline of the study in the introduction.

Response: We will improve the outline of the study in the introduction, and specify earlier the time step (in the abstract and in this outline).

P4.L1 with Rpot being the...

Response: Potential solar radiation. It will be added.

P4.L13 space dependent

Response: It will be modified

P4.L20: reformulate.. a parameterization for the reference parameter for the daily temperature range.

Response: It will be modified

P5.L11. start the paragraph with We want to ..

Response: It will be modified

P5.L12 Fig3b?

Response: Yes, it will be added

P5.L20 ...parameter is proportional to the mean..

Response: It will be modified

P5.L22 Figure 3a?

Response: Fig 3b, it will be added

P5.L23 it appears quite linear to me..

Response: We are speaking here of fig 3b. It will be added to the text.

P5.L26 ..range, and mountain..

Response: It will be modified

P5.L30 the differences in elevation is denoted δ

Response: It will be modified

P6.L4 point instead of area? And the average elevation difference noted.. How did you test?

Response: Yes, "area" should be changed to "point". We tested different descriptors (median of the distribution, range of the distribution, coefficient of variation of the distribution, the skew of the distributions, etc.) with linear and exponential regression compared against $r^2$ values.

P6.L7 If $\delta Ì\check{}E$... and so on for the next lines

Response: It will be modified

P6.L19 which is the average

Response: It will be modified

P6.L21 We do not know that the range depends on...

Response: It was implied that if the radius changes, more pixels are used and the size and the range of the distribution tend to increase. This notion will be added.

P7.L30 reformulate sentence

Response: It will be modified

P8.L18.. humidity is needed.

Response: It will be modified

P8.L24 What happened to Tetens equation?

Response: The equation will be added

P9.L8 Pvap not defined

Response: It will be added

P9.L10 ..point, as humidity is the ..

Response: It will be modified

P9.L15. Be specific about the stations, not "others" and "these"..

Response: The complexity here is that the validation stations are different for SW and LW stations. 18 calibrations stations are used, 15 stations are used for LW validation and 90 for SW validation. We will reformulate and specifically name these different samples in the paper.

P9.L27 mu(mu)???

Response: It will be modified. The hat indicates simulation results

P12.L1 u_prec??

Response: The energy input from the precipitations. It will be added.

P12. Eq. 24,25 no wind speed?

Response: The wind speed is considered constant and is included in the aerodynamic resistance (units in s.m-1).

P12.L25.. is assumed to be negligible compared to other energy fluxes

Response: It will be modified

P15.L3 retro-action again

Response: It will be modified to "feedback"

P15.L17 generality

Response: It will be modified

Figure 6. Eq. 16 does not yield vapor pressure

Response: It should reference the Tetens equation which will be added to the manuscript.

Figure 8 what are the circles. What does the letter indicate? There are no two identical letters.  Do you really need to compare so many models?  We lose track of which is which with the uninformative abbreviations.

Response: The letters indicates the results of the Friedman statistical test. This test compares two identical samples which had different treatments. In this case, the samples are the performances of simulating SW or LW at the same stations using different models (each boxplot is a sample of performances obtained by one specific treatment or model). The Friedman statistical test calculates p-values for each couple of treatments, determining if they are significantly different or not. If two boxplots share the same letter, they are not considered as statistically different (e.g. red boxplots Fig 8a, B2 and B boxplots share the letter b, so they are not significantly different and B and Bs are also not statistically different as they share the letter c). If they do not share a letter, they are considered as statistically different (e.g. red boxplots Fig 8a, B and B1 are statistically different as they do not share any letters). The statistical test can also range the different treatment giving the best performances, this is indicated by the letters, "a" being the best model and "z" being the worst.

The use of these letters will be clarified in the beginning of the result section

Figure 12. Lots of unexplained dots

Response: Dots have been plotted if there are not enough data to use boxplots. The authors apologize as it seems this has been omitted in the figure captions.

Modifications of the structure of the paper:

1 – Introduction

2 – Datasets

3 – Methods framework

   3-1 Parameterization of shortwave radiation

   3-2 Parameterization of Longwave radiations

   3-3 Humidity parameterization

   3-4 The reference surface temperature model

   3-5 The experimental plan

4 – Results

   4-1 Comparison of the different parameterization

   4-2 Comparison of two different calibrating methods

   4-3 Response of a reference surface to the simulated radiative forcing

5 – Discussion

6 – Conclusion

---

## Author Comment (AC2) · 14 Dec 2017

Response to Anonymous Referee #2

We would like to thank the reviewer for his/her interesting comments and suggestions. We are sure that taking these suggestions into account will significantly improve the clarity and the scientific interest of this paper. We try to address them below in a point-by-point format. The original review is presented in black and the response provided is indicated in red.

The authors used data from over 100 weather stations with different aspect and altitude to develop more accurate long and short-wave radiation estimates from readily-available data such as temperature and geographic features. While I believe this to be a worthwhile task, the manuscript as written is hard to understand, and unclear in its major findings. Consequently, I recommend revisions to the paper before it can be published.

First – it was hard to follow all your descriptions of the various parameters used. I'd suggest putting some of the detail in an appendix section, and simply providing the parameter formulations you ended up using in the main documentation.

Response: The authors acknowledge that the paper should be rearranged to improve its clarity. Your suggestion to put some details in an appendix section will be taken into account. Formulations that are not part of the new model we propose will be put in an appendix. In this appendix will be moved the original Bristow formulation (Eqs. 2 & 3), and the $\Delta T_{ref}$ formulations that were used as benchmarks (Eqs. 6, 10 & 11).

Also – I'd like to see clearer comparisons between the new model results and more established methodologies for estimating LW and SW radiation.

Response: In this paper, we already compared the newly developed formulation to the original Bristow and Campbell formulation. It is in our opinion a well-established formulation. If the reviewer meant that LW and SW radiations could be compared to reanalysis data, we provided a detailed answer to this specific issue below.

The use of the KGE criterion is unclear – why not use Pearson correlations directly? If this criterion is indeed superior, please provide explanation of acceptable ranges and max/min values.

Response: The KGE' criterion is somehow similar to the Nash-Sutcliffe efficiency criterion (NSE) as the two criteria are correlated. The range goes from $[-\infty, 1]$, 1 being the best performance that can be achieved; this information will be added to the document.

$$KGE' = 1 - \sqrt{(r-1)^2 + (\omega-1)^2 + (\gamma-1)^2}$$

$$r = \frac{Cov_{so}}{\sigma_s \sigma_o}$$

$$\omega = \frac{\mu_s}{\mu_o}$$

$$\gamma = \frac{\sigma_s/\mu_s}{\sigma_o/\mu_o}$$

Here r is the Pearson correlation criteria (optimum of 1, range $[-1,1]$), $\omega$ is the Bias ratio between observations (subscript "o") and simulations (subscript "s") (optimum at 1, range $[-\infty, +\infty]$) and $\gamma$ is the ratio of the coefficient of variation (optimum at 1, range $[-\infty, +\infty]$). The KGE' is maximized if all subcriteria (r, $\omega$ and $\gamma$) are equal to 1. The KGE' criterion is superior to the Pearson correlation only as it takes also into account the bias and the variation of the variable.

In addition, it would be useful to see how the provided formulations compare to simply using the nearest weather-station values for LW and SW radiation, or reanalysis datasets – i.e provide some more broad comparisons for your methodology.

Response: The objective behind this paper is to create a distributed simulation of SW and LW radiations over the mountain range. Using the nearest weather station value can be used as a kind of validation (Davos and Weissfluhjoch are the only neighboring stations with very different elevations), but cannot be used as an extrapolation method since the number of stations is small for the whole mountain range.

It is of course possible to compare the simulations to the LSASAF remote sensing products providing SW and LW daily simulations at a 3x3 km spatial resolution. Nevertheless, in this case we think it would be difficult to make a good interpretation of the results without using multiple reanalysis datasets using different downscaling (as the space resolution of reanalysis is often coarse), which would increase the complexity of the paper. We could also think of using part of the parameterization developed here for downscaling coarse grid calculated radiation to finer resolution, but we think it is outside of the scope of the present manuscript.

In the end, it didn't appear that your calibrated models performed much better than the original models. For your conclusion - discuss conditions under which using these more complex formulations would be worthwhile.

Response: The simulations of reference surface temperature indeed do not appear to perform much better than the original models. As said in the discussion, it seems that the errors on longwave simulations have a great impact for high elevation areas. We think that even if this work does not solve completely the problem of simulating radiations in mountain areas, the methods and reflections we had can be useful to the community. This paper could be used as a base for improving longwave radiation modeling in mountains environment when reanalysis data are too coarse.

We think that the method we proposed has the advantage of not being site specific, by being able to be used at large scale (even if there is large differences in elevation). We believe that the formulation could be adapted at other mountain ranges only by calibrating the parameters of the model on observed SW and LW data. If not enough data are available, parameters from the radiation model could be conjointly calibrated with snow and/or with hydrological model parameters.

Specific comments:

P8L14: Semicolon needed

Response: It will be added

P9L18: extra "?"

Response: It will be removed

P9L23: Check this equation, it doesn't seem correct in the form written, as it equates three measures of effectiveness similarly in the equation, even though one should be maximized, the others minimized.

Response: The equations have been directly taken from the paper from (Gupta et al., 2009; Kling et al., 2012). The KGE' criterion has to be maximized (maximum value of 1). All the subcriteria from the KGE' also have an optimal performance for a value of 1. If one (or multiple) subcriteria value is inferior or superior to 1, the whole KGE' performance decreases.

Table 4: What are JUN, KOP, and SAM? Please define the acronyms and their significance in the table caption (i.e. do these represent high altitude, mid, and low altitude stations?).

Response: JUN, KOP and SAM are the 3 stations used as examples for further analysis (Figure 4, Figure 10 and Table 4). For the sake of clarity, this information will be added to the table caption.

Modifications of the structure of the paper:

1 – Introduction

2 – Datasets

3 – Methods framework

      3-1 Parameterization of shortwave radiation

      3-2 Parameterization of Longwave radiations

      3-3 Humidity parameterization

      3-4 The reference surface temperature model

      3-5 The experimental plan

4 – Results

      4-1 Comparison of the different parameterization

      4-2 Comparison of two different calibrating methods

      4-3 Response of a reference surface to the simulated radiative forcing

5 – Discussion

6 – Conclusion